# An Integrated Monitoring Approach to the Evaluation of the Environmental Impact of an Inshore Mariculture Plant (Mar Grande of Taranto, Ionian Sea)

**DOI:** 10.3390/biology11040617

**Published:** 2022-04-18

**Authors:** Adriana Giangrande, Margherita Licciano, Daniele Arduini, Jacopo Borghese, Cataldo Pierri, Roberta Trani, Caterina Longo, Antonella Petrocelli, Patrizia Ricci, Giorgio Alabiso, Rosa Anna Cavallo, Maria Immacolata Acquaviva, Marcella Narracci, Loredana Stabili

**Affiliations:** 1Department of Biological and Environmental Sciences and Technologies, University of Salento, 73047 Lecce, Italy; adriana.giangrande@unisalento.it (A.G.); margherita.licciano@unisalento.it (M.L.); daniele.arduini@studenti.unisalento.it (D.A.); jacopo.borghese@unisalento.it (J.B.); loredana.stabili@irsa.cnr.it (L.S.); 2Department of Biology, University of Bari Aldo Moro, 70125 Bari, Italy; cataldo.pierri@uniba.it (C.P.); roberta.trani@uniba.it (R.T.); caterina.longo@uniba.it (C.L.); 3Institute for Water Research, CNR, 74123 Taranto, Italy; patrizia.ricci@irsa.cnr.it (P.R.); giorgio.alabiso@irsa.cnr.it (G.A.); rosanna.cavallo@irsa.cnr.it (R.A.C.); maria.acquaviva@irsa.cnr.it (M.I.A.); marcella.narracci@irsa.cnr.it (M.N.)

**Keywords:** IMTA, mariculture wastes, monitoring program, sediments, water column

## Abstract

**Simple Summary:**

The importance of aquaculture for providing animal proteins to a steeply increasing world population is growing. Despite the many benefits from this practice, there are also many constraints. Among them, the eutrophication of seawater and unsustainability are of the utmost importance. Integrated multitrophic aquaculture (IMTA), consisting of the co-farming of organisms of different trophic levels, was conceived to overcome these problems, and it is still developing to reach sustainable practices. In the Taranto seas (southern Italy, Mediterranean Sea), the first attempt of an IMTA plant with fish, mussels, polychaetes, sponges, and seaweeds started within the framework of the REMEDIA-LIFE project. This plant was arranged in a pre-existing fish farm where the European seabass *Dicentrarchus labrax* and the sea bream *Sparus aurata* are bred in net cages. Here, we report the results of the *ex-ante* evaluation of the trophic conditions of the seawater around the plant, which is useful for assessing the bioremediation effectiveness of the IMTA action after its implementation.

**Abstract:**

The results of an *ex-ante* survey aiming to assess the impact of a fish farm in the Mar Grande of Taranto (southern Italy, Mediterranean Sea) on the surrounding environment are reported. There, the implementation of an innovative IMTA plant was planned, with the goals of environment bioremediation and commercially exploitable biomass production. Analyses were conducted in February and July 2018. Both seawater and sediments were sampled at the four corners of the fish farm to detect the existing biological and physico-chemical features. The investigation was performed to identify the best area of the farming plant for positioning the bioremediating system, but also to obtain a data baseline, to compare to the environmental status after the bioremediating action. Data were also analyzed by canonical analysis of principal coordinates (CAP). All the measurements, in particular, microbiology and macrobenthic community characterization using AZTI’s Marine Biotic Index (AMBI) and the Multivariate-AMBI (M-AMBI) indices, suggest that the effect of fish farm waste was concentrated and limited to a small portion of the investigated area in relation to the direction of the main current. A site named A3, which was found to be the most impacted by the aquaculture activities, especially during the summer season, was chosen to place the bioremediation system.

## 1. Introduction

Mariculture activities, in particular, fish farming, strongly affect the surrounding environment [1]. Wastes deriving from these activities are the main cause of negative environmental impacts causing the eutrophication and organic enrichment of the sediment, influencing water quality and benthic communities [2]. The possible increase in pathogenic bacteria and viruses is also not to be overlooked [3]. In this context, the extension of the mariculture plants, the hydrography of the zone, and the farming techniques play an important role [4,5,6,7]. According to D’Alessandro et al. [8], the measurement of environmental variables can be a direct method to evaluate any possible impact, while the detection of indicator species, especially concerning the benthic habitat, may be an indirect method associated with the first one.

The Trophic Index (TRIX) was designed as a valid tool to assess the trophic conditions of coastal waters [9]. It is based on a logarithmic combination of chlorophyll-a (Chl-a), dissolved inorganic nitrogen (DIN), total phosphorus (TP), and percent oxygen (%O) measurements, and was adopted as the reference index by Italy (D. Lgs 152/99), and then by several other European states [10].

The European Water Framework Directive (WFD; 2000/60/EC) introduced the use of biological quality elements to assess the ecological status of seawater and called on the Member States for the development and intercalibration of biotic indices based on the assessment of species diversity and sensitivity [11]. The AZTI’s Marine Biotic Index (AMBI) based on species sensitivity, was developed to assess the quality of the European coastal and estuarine environments through the evaluation of soft bottom benthic fauna [12]. Successively, the new index Multivariate-AMBI (M-AMBI) was designed to integrate AMBI with species richness and Shannon diversity [13]. These indices also proved to be suitable for the assessment of water quality in fish farming areas [14,15,16,17], and of the recovery of macrobenthic communities from eutrophication [18].

Although the effects of fish farming on sediments and meiofauna have been evaluated in many studies [5,19,20], knowledge of the influence of aquaculture practices on the microbial compartment is still scant [21,22,23]. In a recent study carried out in Brazil to test the effects of chemicals in the treatment of a fish farming plant, the *Salmonella*/microsome assay showed the mutagenic action of fish farm waters [24]. However, microorganisms are of great importance to aquaculture, where they occur naturally or can be unintentionally introduced as contaminants.

Currently, the best strategy to overcome maricultural impact is Integrated Multitrophic Aquaculture (IMTA), which covers the polyculture of different organisms belonging to different trophic levels able to bioremediate the farming plant effluent [3]. This farming technique also has other benefits, such as the possibility to diversify production, by obtaining alternative commercial exploitable biomass to fish [25,26].

The project REMEDIA-LIFE, which is ongoing in the Mar Grande of Taranto (southern Italy, Mediterranean Sea), is a Life project (LIFE16 ENV/IT/000343) aiming to implement an innovative IMTA in a fish farm plant, using sponges, tube-worms, mussels and seaweeds as bioremediating organisms and, at the same time, as sources of biomass commercially exploitable in different industrial fields [27].

The present study represents the *ex-ante* survey conducted within the project in order to determine the environmental status around this fish farm plant. We used an integrated approach with the investigation of both water column and sediment to obtain a detailed description of the environment. This survey was performed not only to plan the position of the bioremediating system within the plant, but also to have reference data relative to the environmental situation existing at the beginning of the project. It represents a baseline which, compared to the results of analysis after the commissioning of the bioremediators, will be useful to check the effectiveness of the innovative IMTA plant in providing environmental sustainability to fish farming activities.

## 2. Materials and Methods

### 2.1. Study Area

The study was carried out in a fishing farm, “Maricoltura del Mar Grande”, located in the Mar Grande of Taranto (40°25′56″ N; 17°14′19″ E) (Ionian Sea) (Figure 1A), at approximately 600 m from the coast. It has a surface area of 0.06 km^2^ and consists of 15 fish cages (Ø 22 m), arranged in three rows each with five cages, working at a depth ranging from 7 to 12 m. Two fish species, i.e., the European seabass *Dicentrarchus labrax* (Linnaeus, 1758) and the sea bream *Sparus aurata* (Linnaeus, 1758), are farmed here, with a yield of approximately 100 tons/year. This mariculture plant participates in the REMEDIA-LIFE project, making six cages available for the innovative IMTA system. In the same zone, the intensive farming of *Mytilus galloprovincialis* (Lamarck, 1819) is also carried out. 

### 2.2. Sampling

Seawater and sediment samples were collected in four stations located around the fish farm (MMG), and labeled A3, A6, B3, and B6, corresponding to the four corners of the area, respectively (Figure 1B). Samples were collected twice during the year 2018 (February and July), corresponding to cold and warm periods. Following the standard sampling methods, seawater samples for nutrient and microbiological analysis were collected using a Niskin bottle. Sediment samples for microbiological investigation were collected by scuba diving and preserved under sterile conditions. The hard bottom community was sampled on permanently immersed artificial hard substrates, such as iron chains and concrete anchoring blocks placed on muddy sediment without vegetation cover. At each station, photographs were acquired and subsequently analyzed with the ImageJ software, annotating the conspicuous fauna species detected and coverage. For a fine identification of the organisms, samples were also collected by scraping off three replicates of 400 cm^2^ at each station. In contrast, soft bottom samples were collected for macrobenthic analysis using an Ekman grab (152 mm^3^), collecting three replicates for each sampling site, for a total of 24 samples, 12 in the February survey and 12 in the July survey. 

### 2.3. Physicochemical Measurements

The hydrodinamics of the area was measured by using the SD6000 current meters of the Sensordata (Vassenden, Norway). Measurements were performed simultaneously, every 5 min for 24 h, in two stations: Stations 1 and station 2 both on the surface (−1 m) and on the bottom (−6 m) with two current-meter chains positioned with a line mooring.

The measurement of the physicochemical parameters was carried out using a multiparameter probe (IDROMAR, IP050D, San Giuliano Milanese, Italy) along the water column up to 12 m, for the evaluation of pH, dissolved oxygen, chlorophyll a, and temperature. The methods used for the analysis of nutrients in water were adapted using the Systea Srl Micromac Lab 1000 multiparameter laboratory analyzer based on a technology called LFA (Loop Flow Analysis) and are based on those published by APAT “Manuals and Guidelines 29/2003 “IRSA-CNR” Analytical methods for water” (ISBN 88-448-0083-7). 

The TRIX index was calculated by using the following formula: TRIX = [log10 (Cha × D% O_2_ × DIN × P) − (−1.5)]/1.2 (1)
where:

Cha = chlorophyll a;

D% O_2_ = % deviation of the oxygen concentration from saturation conditions;

DIN = dissolved inorganic nitrogen;

P = total phosphorus.

For each interval of TRIX values a “Quality Rating” ranging from High to Poor was expressed.

### 2.4. Biological Features

#### 2.4.1. Microbiology

Samples from both compartments (water and sediments) were received in the microbiology laboratory within 4 h of collection, and always stored at 4 °C. The following parameters were determined: total coliforms, fecal coliforms, *Escherichia coli*, fecal enterococci, *Salmonella*, culturable vibrios, culturable heterotrophic bacteria, and bacteria culturable at 37 °C (including potential pathogens). Sediment samples were diluted with filtered (0.22 µm) seawater to obtain a 1:10 (*w*/*v*) dilution and homogenized for 90 s in a sterile Waring blender. The homogenates were then processed in a similar manner to the seawater samples. In order to evaluate the concentration of culturable heterotrophic marine bacteria at 22 °C, 0.1 mL of each sample (seawater or sediment) and appropriate decimal dilutions (10^−1^, 10^−2^, 10^−3^, 10^−4^, and 10^−5^) were plated in triplicate onto Marine Agar 2216 (MA) [29]. After incubation of the plates seeded on MA in the dark at 22 °C for 7 days, the heterotrophic culturable bacteria were counted, according to the colony-forming units (CFU) method [30]. Counts of culturable bacteria growing at 37 °C (including human potential pathogens) in seawater and sediment samples were determined by plating 0.1 mL of each sample and appropriate decimal dilutions (10^−1^, 10^−2^, 10^−3^, 10^−4^, and 10^−5^) in triplicates on Bacto Plate Count Agar (Difco) (BD, Franklin Lakes, NJ, USA) [29]. The plates were incubated at 37 °C for 48 h. To assess the microbial water quality, standard methods (e.g., ISO the International Organization for Standardization) were followed. In particular, total coliforms, fecal coliforms, and fecal enterococci were determined using the most probable number (MPN) method, and using the standard five-tube method of ten-fold dilutions for seawater samples [31]. The coliform bacteria concentration was evaluated by using the miniaturized MPN, in accordance with ISO 9308-3:1998 [32]. Fecal enterococci were measured by using the miniaturized MPN method (incubation at 44 °C for 24–48 h) [33]. Culturable vibrios were enumerated by filtering in three replicates, 1, 5, and 10 mL, of each sample (seawater and sediment) on 0.45 µm pore size filters (Millipore (Merck KGaA, Darmstadt, Germany)). Aseptically, filters were placed onto thiosulphate-citrate-bile-salt-agar (TCBS) plus 2% NaCl, as already reported by [34]. Incubation was performed at 20–25 and 35 °C for 2 days, and the growing colonies of presumptive vibrios were counted in terms of the colony-forming unit (CFU) method.

The incubation temperature of 35 °C was chosen to estimate the fraction of vibrios potentially pathogenic to humans. The lowest incubation temperature (20–25 °C) was selected to detect some *Vibrio* spp., including *V. anguillarum* that do not grow well at 37 °C [35]. After incubation, the colonies of presumptive vibrios (yellow or green), grown on TCBS agar, were counted according to the colony-forming unit (CFU) method. 

The isolation of *Salmonella* in sediments was carried out using the UNI EN ISO 6579:2008 method, and in seawater using the APAT CNR IRSA 7080 procedure; the enumeration of *E. coli* was performed using a five-tube three-dilution MPN method according to APAT CNR IRSA 7020 procedures [36,37]. 

For the acute toxicity of the selected sites, Microtox assay using the *Vibrio fischeri* was employed. In particular, the Microtox^®^ Solid Phase Test (SPT) for sediments and Microtox^®^ Basic Test (BT) for interstitial water were performed according to standard operating procedure [38]. In the SPT sediment, samples were centrifuged at 8000 rpm for 30 min to remove the interstitial water (used in the BT), and then a 7 g (±0.01 g) subsample was tested as a suspension prepared with 35 mL of diluent (Microtox^®^ Solid Phase Test Diluent). Bacteria were exposed to a 1:2 dilution series of the sample and their light output was determined after incubation with *V. fischeri* for 20 min; then, the sample suspension was separated by pressure. The light emission of the bacteria in the pore water was measured after 5 and 15 min and compared to an aqueous control. The tests were performed at 15 °C and pH 8.0 ± 0.5, with two replicates and four controls, according to the standard operating procedure. After the test, each dilution was corrected for turbidity using UV spectrophotometry (Lambda 3B spectrophotometer, Perkin Elmer, Waltham, MA, USA) at 490 nm. In the BT test the interstitial water samples were diluted 1:10 using the diluent reagent (Microtox^®^ Diluent, Modern Water, London, UK). The light output of the bacteria was compared to an aqueous control. The tests were performed at 15 °C and pH 8.0 ± 0.5 with the control [39]. The toxicity assessment of water samples considered the bioluminescence inhibition of *Vibrio fischeri*, while for sediments, the test result was expressed by the STI (sediment toxicity index), allowing us to elaborate specific scales of toxicity [40]. The bacteria (*Vibrio fischeri*) were obtained from AZUR Environmental company (Carlsbad, CA, USA) as freeze-lyophilized cells.

#### 2.4.2. Macrozoobenthos

The seabed under the cages was mainly composed of mud. As already pointed out, the hard bottom macrobenthic community investigated is that growing on artificial hard substrates represented by concrete anchoring blocks and chains for fish cage anchoring.

In both the sampling substrates (hard and soft bottom), all the material was fixed in alcohol, sorted, and identified at the laboratory at the lowest possible taxonomic level. 

To define the state of environmental quality in the four sampled stations, species richness was considered for hard substrates, while for soft bottom substrates, AMBI and M-AMBI indices were applied [13,14].

The observed AMBI and M-AMBI values were derived using AMBI software (version 6.0), which can be downloaded from the freeware http://ambi.azti.es (accessed on 19 September 2018). The AMBI method considers five ecological groups (EG), such as EG I, which includes the most disturbance-sensitive species, while EG V includes the first-order opportunistic species [12]. 

The index was calculated with the following formula:AMBI = [(0 × %EGI) + (1.5 × %EGII) + (3 × %EGIII) + (4.5 × %EGIV) + (6 × %EGV)]/100(2)

Through the obtained value, ranging from 0 to 7, it is possible to classify the water bodies according to the following table (Table 1) proposed by [12]:

M-AMBI integrates the AMBI biotic index, the Shannon–Wiener H’ diversity index and the number of species (S). The value of the M-AMBI varies between 0 (bad) and 1 (high) and corresponds to the ecological quality ratio (EQR). The reference conditions and the type-specific ecological quality ratios for the application of M-AMBI were obtained from the 2018 Regional Agency for Environmental Protection report related to the Apulia region, where the study area is located, and are shown in Table 2.

### 2.5. Multivariate Analysis

Data were analyzed using the canonical analysis of principal coordinates (CAP) performed on the average values of the measured variables and cumulating the two different times, using the software PRIMER (Plymouth, UK) [41]. Analysis was performed considering water and benthic compartments separately.

## 3. Results

### 3.1. Physicochemical Features

Concerning the hydrodynamic measurements, for most of the time during the day and throughout the year, the current was very slow because the area is highly protected from the wind. Measurements in Station 1, indicated a surface current headed south-west, with a maximum flow of 40 m^3^ m^−2^ and a speed of approximately 3 cm s^−1^, while on the bottom, the mean current was directed toward the north-east, with a maximum flow of 29 m^3^ m^−2^ and a speed of approximately 1.4 cm s^−1^. In Station 2, the direction of the surface current was southward, with a maximum flow of 29 m^3^ m^−2^ and with a speed of approximately 1.6 cm s^−1^. On the bottom, the direction of the main current was toward the south-east, with a maximum flow of 21 m^3^ m^−2^ and with a speed of approximately 1.2 cm s^−1^.

At both sampling times, data showed no differences among sample sites in physicochemical values in the water column (Table 3).

The nutrients measured in the water column showed an increase from Site B (B6 and B3) to Site A (A6 and A3) in both the sampling months. The mean values of ammonia ranged from 0.54 μM recorded in July at Station A3 to 0.095 μM observed at Station B3. NO_3_ and NO_2_ had a similar trend, showing highest values in July at Station A3 (approximately 3.94 μM for NO_3_ and 4.6 μM for NO_2_ and lower values in the other stations). The highest total N (N_tot_)values, which is a measure of the nitrogen cycle from land to sea, were recorded in July at Station A3, but high values were observed also at Station B6 during both sampling periods, with values of approximately 50 μM. The PO_4_ mean values were relatively similar across the sampling periods within Stations B3 and B6, with the highest value measured in July at Station A3 (0.23 μM). In contrast, in both the sampling periods, the total P (P_tot_) values, which also include organic phosphorus coming from proteins, phospholipids, and nucleic acids, and are important for monitoring discharges from different sources, such as wastewater treatment plants, were found to be the highest at Station B6 (6.2 μM).

The trophic condition was synthesized using the TRIX index values, as shown in Table 4, in which Station A3 showed the highest value.

### 3.2. Biological Features

#### 3.2.1. Microbiology

In all the analyzed water samples, *Salmonella* spp. was absent (Table 5), while total and fecal coliforms were more concentrated at Stations B3 and B6 in both the sampling periods (Figure 2 and Figure 3).

The highest values were recorded in July corresponding to 280 ± 24 MPN/100 mL for total coliforms, and 130 ± 12 MPN/100 mL for fecal coliforms and *E. coli*. In the sediment samples, the concentrations of total and fecal coliforms were higher at Stations A3 and A6, while the highest density of *E. coli* was detected at Station A3. The highest concentrations were recorded in July with values of 1609 ± 85 MPN/g for total coliforms, and 270 ± 17 MPN/g for fecal coliforms. Additionally, in this case, *Salmonella* spp. was absent. The mean densities of fecal enterococci ranged from 25 ± 1.2 to 9 ± 0.4 MPN/100 mL in the seawater samples with the highest value of 49 MPN/100 mL. In the sediment samples, the mean values ranged from 179 5 ± 75 to 115 ± 10 MPN/g, with the highest value of 3480 MPN/g recorded at Station A3 in July. The mean *Vibrio* density was 5.9± 0.3 × 10^3^ CFU/g in the sediment samples, with the highest value recorded at Station A3 (1.4 ± 0.1 × 10^4^ CFU/g). *Vibrio* concentrations were lower in the seawater samples, with a mean value of 1.7 ± 0.07 × 10^2^ CFU/mL.

The concentrations of these microorganisms were always higher at Station A3 than at the other sites examined in both seawater and sediment samples.

The Microtox assay confirmed that the interstitial water at all stations and in all seasons had no toxic effect. Additionally, the sediments were found to be non-toxic with a STI < 1 (Table 6).

#### 3.2.2. Macrozoobenthos

The analysis of the artificial hard bottom under the cages allowed us to identify a total of 87 invertebrate taxa, 72 of them at the species level. Among them, the more abundant were Ascidiacea (Tunicata) with 25 taxa recorded, and Annelida and Mollusca with 15 and 13 taxa, respectively. Macrobenthos showed a patchy distribution. The analysis of the collected images of standard 20 × 20 cm surfaces showed that at all the sampling stations the macrozoobenthos coverage reached approximately 70% in the cold period, while the warm sampling showed a coverage of approximately 80%. 

The macrozoobenthic assemblages were mainly composed of large filter feeder invertebrates, with a conspicuous presence of the polychaete *Sabella spallanzanii* (Gmelin, 1791) and an occasional presence of the alien calcareous sponge *Paraleucilla magna* Klautau, Monteiro & Borojevic, 2004. Ascidians such as *Microcosmus* spp. Heller, 1877 and *Aplidium ocellatum* Monniot C. & Monniot F., 1987 were also recorded.

In all the areas, the assemblages appeared less diverse in July; moreover, both A stations, especially in this warm period, showed the lowest number of taxa (Figure 4).

Soft bottom communities revealed the presence of 163 taxa, with polychaeta being the most abundant (106 taxa) followed by Crustacea (31 taxa). As regards the species richness (S) and the number of individuals (N), Figure 5 shows a gradient from Station A3 to B6: the first shows poor biodiversity and a high abundance of organisms, while going toward B6, we can see the opposite situation.

Among polychaetes, the families Cirratulidae and Spioniidae were well represented in a number of species, but the most abundant species were *Capitella capitata*, (Fabricius, 1780), *Cirrophorus nikebianchii* Langeneck et al., 2017, *Prionospio malmgreni* Claparède, 1869, *Micronephthys longicornis* (Perejaslavtseva, 1891), *Armandia cirrhosa* Filippi, 1861, *Cirriformia tentaculata* (Montagu, 1808), and *Heteromastus filiformis* (Claparède, 1864). 

The identified communities were typical of confined environments, characterized by a high organic content, as demonstrated by the presence of *C. capitata*, *Malacoceros fuliginosus* (Claparède, 1868), *H. filiformis*, and *Gallardonereis iberica* Martins et al., 2012, but also the mollusk *Varicorbula gibba* (Olivi, 1792), which is an indicator of sediment instability.

As regards the AMBI index, Station A3 showed a value of 5.06 in February and a value of 4.78 in July, corresponding to a heavily disturbed site and a moderate disturbed site, respectively. The remaining stations, A6, B3, and B6, showed lower AMBI values, corresponding to a slightly disturbed site in all the sampling campaigns (Table 7, Figure 6).

A similar trend was enhanced by the M-AMBI index, although the values indicated a relatively better situation at most of the stations (Figure 7).

### 3.3. Multivariate Analysis

Both trophic state and biological parameters were considered for water environments, and only the latter for the benthic environment. The results from water parameter analysis showed B3 as the site that was least subjected to pollution. In contrast A3 was found to be the most impacted site (Figure 8a), with most of the measured parameters located close to the latter station, except for coliforms, which appeared to be more closely related to Station B6, and potential pathogens with A6. A similar trend was also observed in the CAP analysis relative to the sediment samples, but, showing an inverse trend, with potential pathogens more closely linked to Station B6 and coliforms to A6 (Figure 8b).

In both compartments, however, the measured parameters indicated the existence of a clear gradient of impact related to farm production proceeding from the coast (B Zone) which includes Sites B6 and B3, to Sites A6 and A3 (A Zone) suggesting a presumable accumulation of organic matter under the cage located in the A Zone, especially at Site A3. Indeed, Sites A6 and A3 appeared to be characterized by a low environmental quality and, in particular, Station A3 seemed to represent the most impacted site due to the aquaculture activities. In comparison, both B Stations were characterized by a higher environmental quality.

## 4. Discussion

Aquaculture, especially mariculture, represents today the fastest-growing form of food production [42]. However, notwithstanding the benefits that marine aquaculture has brought to society, the negative impact that fish farming can have on the environment is well documented [2,5,6]. Aquaculture activities strongly affect interrelated pelagic and benthic ecosystems with changes occurring in the water, affecting the composition and diversity of the benthic community around the farming cages [15,16,43,44]. The Integrated Multi-Trophic Aquaculture (IMTA) can be a valuable tool toward building a sustainable aquaculture industry [45]. It is a very flexible concept that adds value to plants based on the appropriate choice of organisms with complementary functions within the ecosystem, and with relevant economic value. The benefits of an integrated approach also include the recycling of waste by producing valuable biomass as a by-product, with a reduction of waste release in seawater and of GHG in the atmosphere and the opportunity to farm fish in a suitable environment in a highly effective manner [45,46,47]. An accurate knowledge of the area is, however, needed before beginning this integrated approach.

Within a life project (REMEDIA-LIFE), we planned a system in an aquaculture facility located along the coast of Apulia, utilizing some organisms for the first time at the preindustrial level. The investigated area is in a relatively confined environment that does not allow the high dilution and dispersion of the waste far from the cages, so that the organic load also remains confined.

Before beginning bioremediating action, as hypothesized in the project, we needed to determine the environmental situation existing below and around the cages. Therefore, we conducted the above-described monitoring survey to obtain a baseline to compare throughout the project. Only through the analysis of the results of this monitoring will it be possible to quantify the differences in the environmental quality occurring after the action of the selected bioremediators, which in turn will be a proof of the success of bioremediation. In addition, the assessment of the ecological status was also necessary to individuate the area where the bioremediators could be placed.

Concerning the trophic state of the water column, the values reached in the different sampling zones near the experimental fish farm in Taranto appeared to be clearly different, with the highest values recorded in A3. However, they also showed that the quality level of the zone was “high-good” except at Station A3, where the mean TRIX value of 4.9 was calculated. The TRIX index ranges from 2 to 8, with four quality classes from “high” to “poor” [10], and values typical of eutrophic areas, were present for example in the northern Adriatic (TRIX = 6.03) [10]. This index is widely used for the assessment of the trophic pattern in coastal waters; it has been rarely applied to monitor the water trophic level in fish farms [48,49]. The observed values at Station A3 are comparable to those calculated in a fish farm area in the Trieste Gulf, where TRIX ranges from 4.86 to 5.05 [48], while those found in the Izmir Bay (Turkey) approximately 3.6 [50], were comparable with the other stations. 

As regards the microbiological analyses, it is noteworthy that at present in Europe the only regulatory and legal constraints in aquaculture policies regarding the quality of waters concern mainly shellfish. The main reason for this is that aquaculture is not recognized as a user of water resources in the same way as other users, such as fishing or tourism. Thus, so far, the EU’s water policy is governed by two instruments: the Water Framework Directive, which covers inland and coastal waters, and the Marine Strategy Framework Directive 2008/56/EC, which covers marine waters. In this scenario, in order to discuss our data, we referred to the current Italian and EU regulations concerning the assessment of farming localities for the cultivation of bivalves. *Salmonella* spp. has never been detected in the waters of the aquaculture farm considered here. Furthermore, all the recorded values of *E. coli* and the other examined microbial pollution indicators were found to be lower than the legal limits imposed by the above mentioned national and European regulations, even in July when the densities of several microbiological parameters increased. It is remarkable that enterococci are not cited in the current regulations for mussel culture. However, taking into account that coliforms and enterococci are often used as indicators of the fecal contamination of potable and recreational water, as well as food, in the present investigation, we broadened the spectrum of microbiological analyses and evaluated both coliforms and enterococci. Furthermore, enterococci represent an indicator of older fecal contamination as they survive better in the environment [51].

The heterotrophic bacteria densities recorded in the present work were similar to those already measured in other Mediterranean areas, where mariculture is practiced [52]. In particular, fluctuations in abundance, with higher values in July than in February, were observed for heterotrophic bacteria and vibrios, which were correlated with the ecology of these microorganisms. The warm temperature and the high availability of nutrients, increased their growth [53]. In general, it has been found that in aquaculture systems, the main driver of the bacterial community among all environmental factors is temperature [54,55]. Therefore, outbreaks of aquatic animal disease in aquaculture are often seasonal and could be closely related to seasonal changes in microbial populations [56]. Bacteria belonging to the genus *Vibrio* are of particular concern, as they represent a considerable portion of the halophilic marine bacterial populations, are highly thermally dependent, and often are associated with human as well as marine animals’ diseases, called vibriosis. 

Although the two compartments showed different concentration of these main microbiological groups, the Microtox results showed no toxicity both for sediments and interstitial water in all the samples examined. This is in accordance with the studies of [57] who employed *V. fischeri* to assess the toxicity of sediments surrounding fish farms. However, the densities of the other examined microbiological parameters were higher in the sediment samples than in the water ones. In particular, culturable heterotrophic bacteria, *E. coli*, fecal enterococci and vibrios reached high densities in the sediment samples at Station A3.

Macrobenthic investigation also indicated A3 as the highest disturbed station, with no significant differences between the two seasons. Here, the community, typical of confined environments, was dominated by the presence of the species *C. capitata*, emphasizing the high organic enrichment present in this site compared to the other analyzed stations. A similar result was obtained using M-AMBI index, showing that Site A3 had the lowest score, indicating a moderate EQS, while Site B6 had the highest score. These results appeared to be coherent with hydrodynamic measurements; the north east direction of the main current, measured at the bottom, suggested that organic pollution from the plant was directed toward Site A3. Indeed, AMBI and M-AMBI highlighted a sort of gradient of environmental quality going from Station A3 to Station B6, which was particularly evident in February. The gradient was determined by decreasing AMBI values and increasing M-AMBI values, coupled with a decrease in the percentage of opportunistic species and increasing richness and diversity, and the presence of sensitive species. Several of the measured parameters, therefore, indicated the existence of a clear gradient of impact related to farm production proceeding from the coast (B Zone) to the A Zone, suggesting a presumable accumulation of organic matter under the cage located at Station A3. Indeed, from the results presented here, the Sites A6 and A3 were characterized by a low environmental quality and, in particular, Station A3, especially in July, seemed to represent the most impacted site due to aquaculture activities. In comparison, both B Stations were characterized by a higher environmental quality. This trend was highlighted, especially considering the sediment compartment, in particular, microbial density and macrobenthic organism distribution, both from hard and soft bottoms.

Thus, results from this study highlighted a relatively local impact of the farm activity on the sediments, which was limited to the area close to Station A3.

This situation was highlighted by the multivariate analysis, where the water column and benthic parameters showed similar results especially, considering Stations A3 and B3, for which water column parameters indicated A3 as the more eutrophic one, while sediment analysis indicated A3 as the more disturbed benthic area. Few differences can be enhanced when comparing the models obtained by the analysis of the water column and the sediments, which could be due to the current going in opposite direction on the surface and over the sediment. However, it must be underlined that the benthic compartment can be considered the biological memory of the system, especially for the length of the life cycle of the organisms and for the close relationship that invertebrates have with the sediment, so that they appear particularly sensitive and indicative [42,58].

Overall, our finding indicated Site A3 as the most impacted from the aquaculture activities; therefore, this site was chosen to place the bioremediation system.

## 5. Conclusions

The *ex-ante* analysis conducted in the small mariculture plant Maricoltura Mar Grande of Taranto, where the REMEDIA-LIFE project is still ongoing, revealed how the impact of the plant is quite limited and circumscribed to a small area of the plant, which will be taken into consideration to measure the effects of the bioremediation systems we planned. Since the project was conceived to contain the environmental impact of the breeding cages in a semi-enclosed area, we needed to understand how to design the bioremediation system and to have a picture of the situation before the bioremediating intervention. All the examined biological parameters gave the same indication and represent a baseline for further comparison. The monitoring action allowed us, in fact, not only to understand where the wasted accumulated, but represents the starting situation for comparison with the data obtained after the working of the bioremediating system. The survey, in fact, will be repeated at the same sites for several years and in the same periods of the year in order to gain an idea of the changes that may be due to the treatment of the site with the bioremediating system.

## Figures and Tables

**Figure 1 biology-11-00617-f001:**
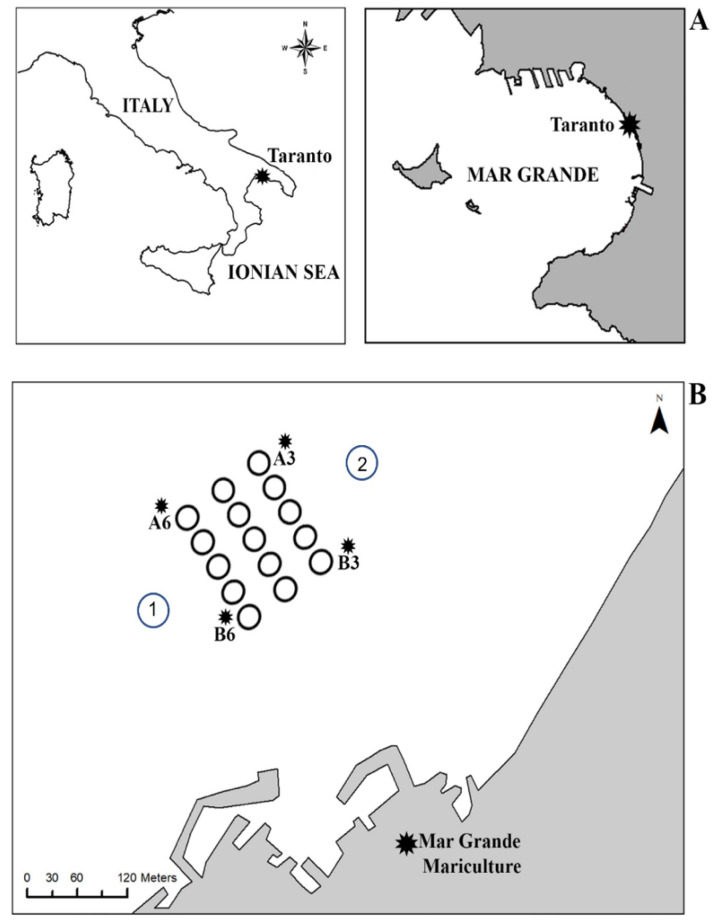
Map of the study area in the Mar Grande of Taranto (**A**). A3, A6, B3, B6: sampling sites relative to both water column and sediments (**B**). Stations 1 and 2 refer to the site where current measurements were taken (from [28] with permission from IEEE, 2022, Copyright Clearance Center’s RightsLink^®^).

**Figure 2 biology-11-00617-f002:**
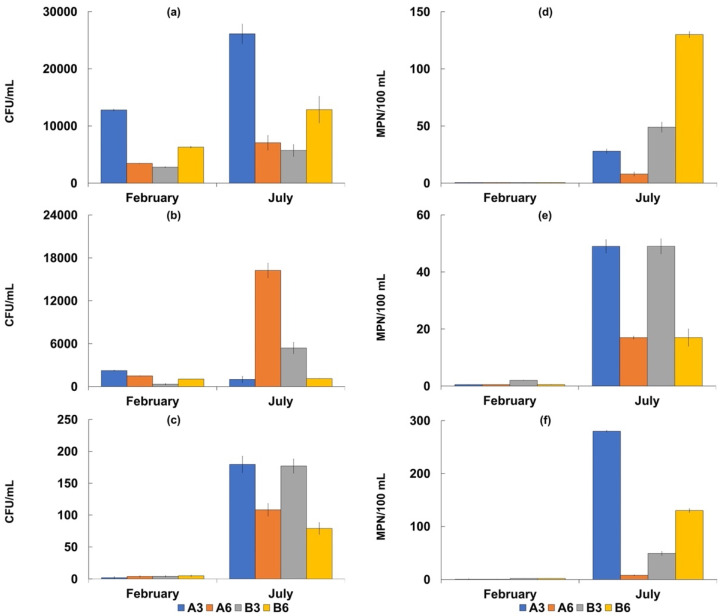
Concentrations of the considered microbiological parameters in seawater samples collected in February and July 2018 at the four sampling sites (A3, A6, B3 and B6). Bacterial counts are reported as mean values ± S.D. of three replicates. Bacterial counts are expressed as CFU/mL for culturable heterotrophic bacteria at 22 °C (**a**), culturable bacteria at 37 °C (**b**) and culturable vibrios (**c**), and as MPN/100 mL for fecal coliforms (**d**), fecal enterococci (**e**), and total coliforms (**f**).

**Figure 3 biology-11-00617-f003:**
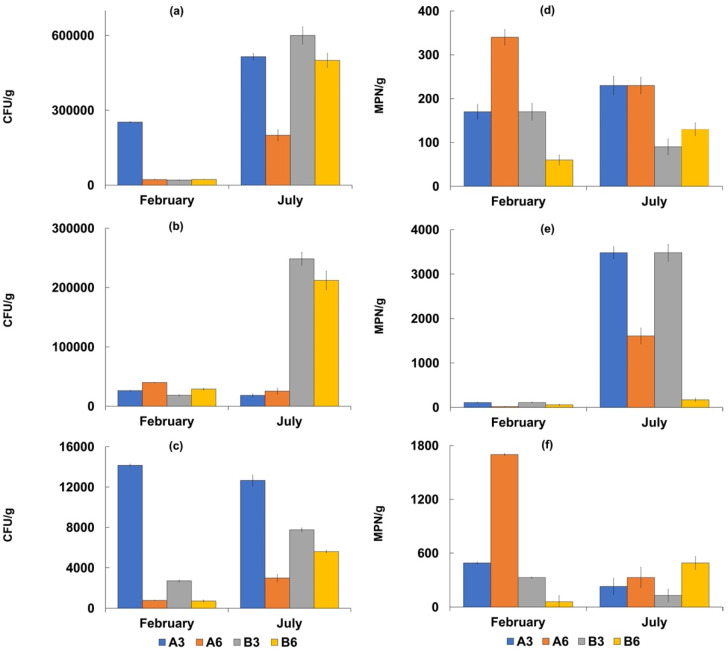
Concentrations of the considered microbiological parameters in sediment samples collected in February and July 2018 at the four sampling sites (A3, A6, B3, and B6). Bacterial counts are reported as mean values ± S.D. of three replicates. Bacterial counts are expressed as CFU/g for culturable heterotrophic bacteria at 22 °C (**a**), culturable bacteria at 37 °C (**b**) and culturable vibrios (**c**), and as MPN/g for fecal coliforms (**d**), fecal enterococci (**e**) and total coliforms (**f**).

**Figure 4 biology-11-00617-f004:**
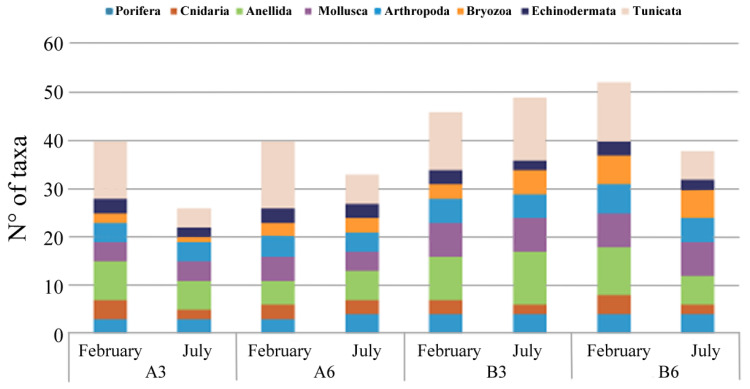
Hard substrate taxa abundance detected at each sampling site (A3, A6, B3, and B6) in February and July 2018.

**Figure 5 biology-11-00617-f005:**
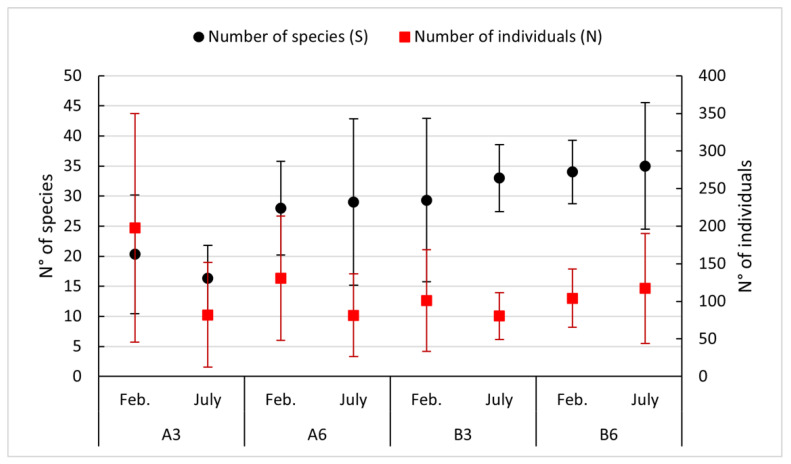
Trends of abundance and species richness in the study sites in the year 2018.

**Figure 6 biology-11-00617-f006:**
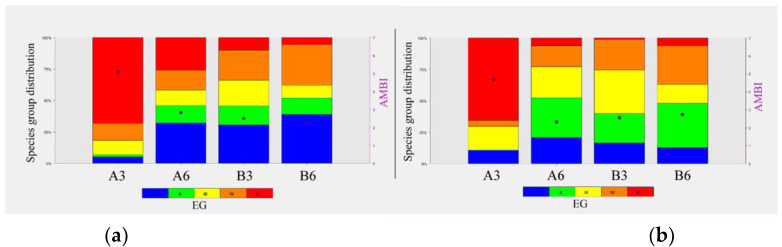
AMBI values and representation of the species group distribution at the different stations in February 2018 (**a**) and July 2018 (**b**) campaigns.

**Figure 7 biology-11-00617-f007:**
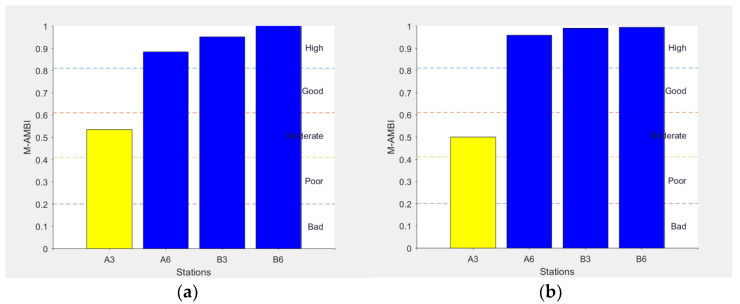
M-AMBI values and respective ecological status relative to the 4 examined sites in February (**a**) and July (**b**) (modified from [28] with permission from IEEE, 2022, Copyright Clearance Center’s RightsLink^®^).

**Figure 8 biology-11-00617-f008:**
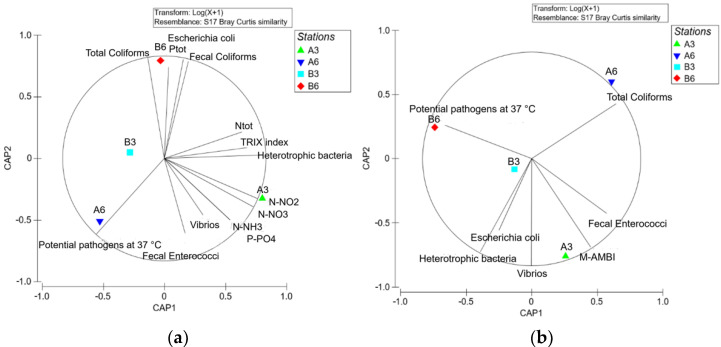
CAP analysis of the mean (February and July 2018) values of the measured parameters: (**a**) water sample; (**b**) sediment samples (modified from [28] with permission from IEEE, 2022, Copyright Clearance Center’s RightsLink^®^).

**Table 1 biology-11-00617-t001:** AMBI values and relative water body classification. Key: AZTI’s Marine Biotic Index (AMBI).

AMBI Values	Water Body Classification
0 < AMBI ≤ 1.2	High
1.2 < AMBI ≤ 3.3	Good
3.3 < AMBI ≤ 4.3	Moderate
4.3 < AMBI ≤ 5.5	Poor
5.5 < AMBI ≤ 7	Bad

**Table 2 biology-11-00617-t002:** The reference conditions and the type specific EQR. Key: AZTI’s Marine Biotic Index (AMBI); Shannon–Wiener diversity index (H’); N° of species (S).

Reference Conditions			EQR	
AMBI	H’	S	High/Good	Good/Moderate
0.5	4.8	50	0.81	0.61

**Table 3 biology-11-00617-t003:** Physical and chemical parameters measured in February and July 2018 (from [28] with permission from IEEE, 2022, Copyright Clearance Center’s RightsLink^®^).

FEBRUARY 2018				
	Stations			
Parameters	A3	A6	B3	B6
Temperature C°	17.22 ± 0.93	17.44 ± 1.00	17.26 ± 0.80	17.50 ± 0.92
pH	7.08 ± 0.07	7.05 ± 0.05	7.11 ± 0.04	6.97 ± 0.08
Turbidity FTU	12.66 ± 0.01	12.66 ± 0.02	12.65 ± 0.03	12.66 ± 0.01
Salinity PSU	38.75 ± 0.23	38.76 ± 0.24	38.78 ± 0.22	38.66 ± 0.22
**JULY 2018**				
	**Stations**			
**Parameters**	**A3**	**A6**	**B3**	**B6**
Temperature C°	26.00 ± 0.67	25.88 ± 0.32	25.84 ± 0.39	25.89 ± 0.35
pH	7.65 ± 0.05	7.65 ± 0.02	7.65 ± 0.02	7.64 ± 0.03
Turbidity FTU	12.62 ± 0.02	12.63 ± 0.03	12.61 ± 0.01	12.60 ± 0.01
Salinity PSU	38.01 ± 2.93	38.32 ± 0.02	38.35 ± 0.35	38.34 ± 0.02

**Table 4 biology-11-00617-t004:** TRIX index values.

	TRIX Index	
Station	February 2018	July 2018
A3	4.49	5.36
A6	3.02	3.53
B3	2.01	2.91
B6	2.84	4.31

**Table 5 biology-11-00617-t005:** *Escherichia coli* and *Salmonella* spp. results (+ presence; −absence).

FEBRUARY 2018						
	Water			Sediment		
	*Escherichia coli*		*Salmonella* spp.	*Escherichia coli*		*Salmonella* spp.
Samples	MPN/100 mL	95% Confidence Interval	+/−	MPN/g	95% Confidence Interval	+/−
A3	2	-	-	270	90–800	-
A6	4	<0.5–13	-	70	10–170	-
B3	2	<0.5–7	-	7	1–17	-
B6	2	<0.5–7	-	170	<50–460	-
**JULY 2018**						
	**Water**			**Sediment**		
	** *Escherichia coli* **		** *Salmonella* ** **spp.**	** *Escherichia coli* **		** *Salmonella* ** **spp.**
**Samples**	**MPN/100 mL**	**95% Confidence Interval**	**+/−**	**MPN/g**	**95% Confidence Interval**	**+/−**
A3	33	11–93	-	40	<5–130	-
A6	7	1–17	-	40	<5–130	-
B3	27	9–80	-	40	<5–130	-
B6	130	35–300	-	20	<5–70	-

**Table 6 biology-11-00617-t006:** Microtox bioassay results.

	FEBRUARY 2018		JULY 2018	
	Sediment	Interstitial Water	Sediment	Interstitial Water
Samples	STI	% Bioluminescence Inhibition	STI	% Bioluminescence Inhibition
**A3**	0.07	Hormesis	0.33	Hormesis
**A6**	0.01	Hormesis	0.01	Hormesis
**B3**	0.02	Hormesis	0.13	Hormesis
**B6**	0.02	Hormesis	0.04	Hormesis

**Table 7 biology-11-00617-t007:** Biotic indices values. Key: station (St.), AZTI’s Marine Biotic Index (AMBI), Multivariate-AMBI (M-AMBI), Ecological Quality Ratio (EQR).

FEBRUARY 2018											
St.	I(%)	II(%)	III(%)	IV(%)	V(%)	Richness	Diversity	Mean AMBI	Disturbance	M-AMBI	EQS
A3	5.2	1.7	11.3	13.5	68.2	43	2.44	5.06	Heavily disturbed	0.52	Moderate
A6	32.1	13.9	12.3	15.7	26	57	4.28	2.84	Slightly disturbed	0.86	High
B3	30.5	15.1	20.5	23.8	10.1	60	4.91	2.51	Slightly disturbed	0.94	High
B6	39	13	10.1	32.5	5.5	67	4.89	2.28	Slightly disturbed	0.99	High
**JULY 2018**											
**St.**	**I(%)**	**II(%)**	**III(%)**	**IV(%)**	**V(%)**	**Richness**	**Diversity**	**Mean AMBI**	**Disturbance**	**M-AMBI**	**EQS**
A3	8.3	1.7	19.2	4.2	66.7	30	2.31	4.78	Moderately disturbed	0.46	Moderate
A6	20.5	28.3	25.4	19.7	6.1	53	5.09	2.44	Slightly disturbed	0.91	High
B3	14.3	29.6	33	21.7	1.3	56	5.05	2.49	Slightly disturbed	0.93	High
B6	12.9	38.7	14.9	27.2	6.3	66	5.17	2.63	Slightly disturbed	0.98	High

## Data Availability

Not applicable.

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
