# Peer review of "An Integrated Monitoring Approach to the Evaluation of the Environmental Impact of an Inshore Mariculture Plant (Mar Grande of Taranto, Ionian Sea)"

_biology, 2022, doi:10.3390/biology11040617_

Round 1
Reviewer 1 Report
The manuscript could be considered for publication after that the comments in the marked attached manuscript will have been carefully addressed.

Reviewer 2 Report
Giangrande et al
This manuscript reports a range of measurements in water and sediment around a fish farm in Mar Grande, Taranto.
1 Data quality: There is no evidence of the reliability or accuracy of the data presented. There is no information on calibration of instruments and no information on the analyses of reference materials.
Some of the data appear doubtful, for example:
- a) Table 3: The uncertainty in temperature measurements of up to a degree seems very large.
- b) The uncertainty in Salinity for A3 in July is huge and implies a salinity range of up to almost 41 psu, which seems highly unlikely.
- c) The turbidity is rather high compared to other published data for Mar Grande, and also highly invariant, which seems unlikely.
I therefore have low confidence in the data presented.
2 Fig 4. The trend lines, which link across both locations and time, are meaningless. The Figure should be deleted.
3 Sampling dates: There is no evidence that sampling on two occasions in a year gives a true picture of the variability within the year.
4 No information is given on the stocking levels or feeding rates at the farm, both of which are likely to affect environmental conditions.
5 Critically, there are no control sites in the sampling design. It is therefore not possible to ascribe any differences in environmental conditions to the aquaculture activity. This is a fatal flaw in the experimental design. For this reason alone, the manuscript should be rejected.
Reviewer 3 Report
The manuscript includes the results of an ex-ante evaluation of the conditions of the seawater and sediments around an aquaculture plant where an Integrated multitrophic aquaculture (IMTA) is projected, also to determine the best place to locate a bioremediation system.
The whole manuscript is very well written and referenced.
I consider that a topic of great interest is being addressed, taking into account the social, economic an environmental implications of a growing aquaculture industry and the importance of implementing an integrated approach for the development of aquaculture projects through which impacts are minimized while production is diversified, reducing the cost/benefit ratio.
I have included some recommendations in the document that I believe will contribute positively to it.

Round 2
Reviewer 1 Report
I acknowledge the efforts done by the Authors to improve the original manuscript. The study could be accepted after minor revisions (see my comments in the attached marked manuscript).

Reviewer 2 Report
I do not see that the authors have addressed the lack of information on the performance characteristics of the measurement methods, or of the reliability of the results.
The authors have not adequately addressed the lack of appropriate experimental design, namely the lack of control site data. They note the difficulty of using a remote site in a BACI design, which may well be the case. However, this does not remove the need for a robust experimental design.
I therefore conclude that major failings of the work have not be resolved and that the manuscript should be rejected.
